# Interplay between Networking Capability and Hospital Performance in Indonesia’s Medical Tourism Sector

**DOI:** 10.3390/ijerph20010374

**Published:** 2022-12-26

**Authors:** Muhtosim Arief, Mohammad Hamsal, Sri Bramantoro Abdinagoro

**Affiliations:** Research in Management, Doctorate Program, BINUS Business School, Bina Nusantara University, Jakarta 11480, Indonesia

**Keywords:** digital transformation, hospital performance, medical tourism, medical tourism ecosystem, networking capability, resources integration

## Abstract

Medical tourism is an industry that is currently developing, but research that focuses on the analysis of supply and institutions as a unit in Indonesia is sparse. This research aims to investigate the variables of digital transformation and resource integration that influence hospital performance, considering the expansion of hospitals’ capacities for networking in medical tourism. A structural equation model is used to evaluate the hypothesis. According to our findings, digital transformation and resource integration both have a beneficial effect on networking capabilities. On the other hand, the implementation of digital transformation does not improve hospital performance. Digital transformation has a good link with hospital performance but has no statistically significant effect on hospital performance. In the meantime, the capabilities of resource integration and networking have a positive effect on the overall operation of hospitals. In a similar vein, the ecosystem of medical tourism helps to improve the connection between a hospital’s networking capabilities and its overall performance. It is anticipated that the findings of this study will serve as a reference for those in the policymaking and healthcare industries to develop medical tourism.

## 1. Introduction

Medical tourism refers to the practice of crossing international boundaries to receive medical care [1]. Medical tourism encompasses not only overseas travel, but also travel inside the same country, with two goals in mind: to take advantage of affordable, high-quality health care services and to have a pleasant trip [2,3]. Regulation No. 76 of the Ministry of Health of the Republic of Indonesia, outlined in 2015, defines medical tourism as an out-of-town or international visit to a hospital for an examination, medical treatment, or other health-related services. Medical tourism is described as traveling both locally and internationally to receive medical care.

According to the Medical Tourism Association, fourteen million individuals travel annually for medical treatment, with an estimated USD 125 billion spent in 2021 [4]. Oguz et al. [5] documented the incomes and number of medical tourists visiting five countries in 2019. Thailand was the nation with the biggest number of medical tourism travelers, with 3 million, followed by Mexico with 1 million, Turkey with 850,000, and India with 551,000 [5].

The International Healthcare Research Center produced an index of medical tourism destination countries according to three criteria: the destination environment, the medical tourism sector, and the quality of services and facilities offered [4]. The average index for Asia is the highest (69.06), followed by Europe (67.04), North America (65.89), Africa (63.80), and the Middle East (62.5). There are 46 nations that have become destinations for medical tourism, including three from Southeast Asia: Singapore, Thailand, and the Philippines, which are ranked second, third, and fourth, respectively [4].

Contrary to popular belief, Indonesia is not one of the world’s main medical tourism destinations; rather, many individuals utilize the facilities of neighboring countries. Indonesia is also not included in the medical tourism destination index compiled by the IHRC [4]. According to the Ministry of Health of the Republic of Indonesia [6], 600,000 to 1,000,000 Indonesians seek medical care overseas at a cost of USD 10,3 billion. Trust, reputation, previous negative experiences, quality of care, breadth of services, and cost of treatment are among the reasons why Indonesians seek health care abroad [7]. Cosmetic and reconstruction surgery, oncology, orthopedics, dental therapy, spinal surgery, ophthalmology, weight-loss surgery, and cardiology are the most sought-after medical tourism treatments [7].

With 2945 hospitals, 80.9% of which are accredited [8], Indonesia has the potential to expand its medical tourism industry. Up to 54 class A hospitals and 403 class B hospitals are accredited hospitals [8]. Notably, 5 hospitals in Sumatra (18.5%), 18 hospitals in Java (66.7%), 2 hospitals in Bali (7.4%), and 2 hospitals in Sulawesi (7.4%) are among the 27 hospitals in Indonesia that have been accredited by an international accreditation agency, notably the Joint Commission International [9].

According to Tham [3], domestic medical tourism can expand with the participation of numerous stakeholders. The growth of high-quality medical tourism must be fostered by health service providers, and the networking capabilities of tourists and intermediaries must improve [10] because networking ability impacts organizational performance [11,12,13]. Stakeholders in medical tourism must work together to develop policies that deliver tourist-centered medical care for optimal healthcare outcomes [14]. The capacity to network with a variety of parties will create a competitive edge and enhance hospital performance.

Through digital transformation, organizations can obtain a competitive edge. As stated by Morakanye et al. [15], the establishment of competitive advantage is one of the outcomes of digital transformation. Organizations must undertake digital transformations in response to the digital revolution’s effects on digital technology, digital competitiveness, and consumer behavior [16]. As medical tourism involves going from one area to another, digital access to information is required. A respondent’s decision to engage in medical tourism is influenced by the time taken to gather information about health services [17]. Additionally, telehealth and telemedicine services encourage medical tourism [18].

The healthcare industry, including hospitals, is undergoing fast development and is experiencing obstacles to achieving corporate goals [19]. The unanticipated COVID-19 epidemic resulted in tremendous economic, human, and social calamity [20]. The literature shows how systems [21], organizations [22], and people [23] respond to, recover from, and adapt to crisis situations. For a successful response to the COVID-19 problem, a robust healthcare system with active participation at the individual, organizational, and government levels is necessary [24].

There are three tiers of stakeholders within the ecosystem of medical tourism in Indonesia. Hospitals, hospital associations, physicians, physician associations, healthcare platforms, and insurance make up the first tier. The second level is made up of travel and tourism providers, which consist of transportation services, airlines, and travel businesses; natural, cultural, and artificial tourism; financial and banking services, housing; tour guides, tour operators, and travel agents; and restaurants. The third level, public and government institutions, consists of the Ministries of Health, tourism and creative economy, KADIN, universities, BKPM, and municipal governments [25].

Indonesia has many hospitals that have been accredited both nationally and internationally. Tourist destinations in Indonesia are no less interesting than neighboring countries that have made it into the international medical tourism index, such as Singapore and Thailand. Indonesia already has two main industrial assets needed for medical tourism, namely, quality hospitals and leading tourist destinations. However, in practice, Indonesians who travel overseas for medical care spend USD 11,5 billion annually [6]. Therefore, Indonesia needs to make efforts to develop its medical tourism industry.

Until now, it is hoped that the government and stakeholders would be able to determine what Indonesian hospitals must do to prevent people from seeking treatment overseas, while simultaneously inviting foreigners to travel to Indonesia for medical care. This necessitates in-depth research that systematically monitors the factors associated with the global development of medical tourism, including the nations that have effectively produced these products. This study intends to assess the influence of digital transformation and resources integration on hospital performance, as mediated by networking capabilities, given the foregoing context. In addition, the medical tourism ecosystem is utilized as a moderating variable in this study. The outcomes of this study are anticipated to serve as a guide for policymakers and the health industry as they develop their services in the context of tourism.

## 2. Materials

### 2.1. Networking Capability

Networking capability is a firm’s ability to exploit existing ties and explore new ties to reconfigure resources and achieve competitive advantage [26,27,28]. Mu et al. [29] and Mu and Di Benedetto [30] define networking capabilities as a company’s competency to seek and find network partners, as well as manage and utilize network relationships to create value. Arasti et al. [31] explain that networking capabilities are a series of activities and routines between companies to develop and manage networks by accessing and integrating resources for value creation.

Networking skills in health services are needed to improve organizational performance. Previous research shows that there is a positive relationship between networking skills and organizational performance [11,12,13]. In the development of medical tourism, Hyder et al. [32] stated that trust and the ability to build networks are needed to reduce unfavorable characteristics, instability, and lack of legitimacy caused by institutional constraints in emerging markets.

### 2.2. Stakeholder Theory

Ireland et al. [33] define stakeholders as individuals who are affected by company performance and who have claims on the company’s performance. Meanwhile, Bonnafous-Boucher and Rendtroff [34] state that stakeholders can be influenced by companies without being able to influence them (and vice versa). Stakeholders are in a network called an ecosystem. An ecosystem is a group of companies that interact and depend on one another’s activities [35]. Vargo and Lusch [36] define the healthcare ecosystem as an independent and adaptable system of actors that integrates resources for shared value creation.

The development of medical tourism provided by hospitals requires stakeholder involvement. Tham [3] said that domestic medical tourism can develop if there is involvement from stakeholders. Stakeholders in the medical tourism industry include retailers, insurance, legal services, banking, universities, government, tourism agencies, product suppliers, and professional associations [37]. Kamassi et al. [38] state that stakeholders in the medical tourism industry consist of medical tourists, health service providers, government agencies, facilitators, accreditation and credentialing bodies, health service marketers, insurance providers, as well as infrastructure and facilities. Kamassi et al. (2020) also stated that the success of developing medical tourism depends on the cooperation of all stakeholders.

### 2.3. Digital Transformation and Hospital Performance

Digital transformation favorably impacts innovation and business performance [39]. Using Tobin’s q (a measurement of financial market-based company performance) and evaluating the relationship between information technology investment and firm q value, information technology contributes to the prospective future performance of firms [40]. Technological capabilities of the company had a significant impact on its performance [41,42]. The adoption and application of information and communication technologies [43] and big data analytic capabilities [44,45] also have a favorable impact on firm performance, according to additional studies. Technology advancement has led to the enhancement of business performance [46]. The findings of this study demonstrate a correlation between the utilization of information technology and the performance of a corporation. This paper formulates the following hypothesis:

**H1:** *Digital transformation has a positive effect on hospital performance*.

### 2.4. Digital Transformation and Networking Capability

Vesalainen and Halaka [47] postulated a bidirectional relationship between networking capability and technological capability. First, the resources and activities associated with networking capability act as an element of the capability set. The parallel appearance and interconnectedness of technological and networking capabilities at the core capabilities level can be observed in a number of situations explored by [47]. Information technology maturity has a significant effect on network visioning capability, network constructing capability, and network centering capability [48]. Cenamor et al. [49] also stated that digital platform capability has a significant positive effect on networking capability. On the basis of this research, this paper formulates the following hypothesis:

**H2:** *Digital transformation has a positive effect on networking capability*.

### 2.5. Networking Capability and Hospital Performance

There is a correlation between networking ability and organizational success [11,12,13]. Absorptive capacity, one aspect of networking capability, influences hospital performance [50]. Networking capability increases the connection between digital capabilities and enhanced business performance [51]. Networking capabilities influence corporate performance positively [52]. The authors of this paper formulated the following hypothesis based on past research findings:

**H3:** *Networking capability has a positive effect on hospital performance*.

### 2.6. Resource Integration and Hospital Performance

A company’s performance depends on how it generates, utilizes, and configures its resources [53]. The resources-based perspective also identifies strategic resources as a characteristic of corporate performance [54]. Barney and Clark [55] argue that a company’s performance can be enhanced by focusing on its resources and paying close attention to their qualities. Company performance can also be forecasted based on the company’s assets [56] and competencies [57]. Meng and Yang [58] also stated that resource integration can boost corporate performance. Previous research indicates that a company’s performance is affected by its ability to integrate its resources. Consequently, this study develops the following hypothesis:

**H4:** *Resource integration has a positive effect on hospital performance*.

### 2.7. Resource Integration and Networking Capability

The definition of a company’s capability is the capacity to “organize, administer, coordinate, or govern sets of operations” [59]. There are two types of firm capabilities in relation to resources: the resource itself and the activity that uses the resources [47]. The existence of tangible and intangible resources is one indication of COMP Ltd. company’s customer-focused networking capabilities [47]. Networking capabilities have always been linked to resource sharing between business partners [60]. Additionally, prior research suggests that networking capabilities may be the optimal method for organizations with limited resources to meet resource requirements and share business risks [60,61,62]. As networking capability is one of a company’s capabilities requiring effective resource integration, researchers construct the following hypothesis:

**H5:** *Resource integration has a positive effect on networking ability*.

### 2.8. Medical Tourism Ecosystem, Networking Capability, and Hospital Performance

According to Bulatovic and Iankova [10], governments, health service providers, and tourist organizers must work together to grow medical tourism. Lack of cooperation between medical tourism facilitators and doctor’s offices is one of the obstacles to the development of medical tourism [63]. Actors in the business ecosystem can actively and passively introduce and utilize diverse sorts of resources to produce value for themselves and other ecosystem actors [24]. According to research on medical tourism in Malaysia, most cooperation between health care providers and tourism companies in Malaysia are also informal, despite the fact that they want to exploit medical tourism for commercial expansion [64]. The development of medical tourism requires strong cooperation amongst all medical tourism ecosystem stakeholders [38]. According to the previous study’s conclusions, hospital performance will not improve upon growing medical tourism if there is no supporting ecosystem and its organizers lack networking skills. Based on the findings of prior experiments, the authors of this paper generated the following hypothesis:

**H6:** *A medical tourism ecosystem strengthens the relationship between networking capability and hospital performance*.

### 2.9. Digital Transformation, Hospital Performance, and Networking Capability

Networking capabilities are crucial in strengthening a company’s competitive edge, especially as high-tech businesses in China continue to search for ways to increase their competitive advantage other than imported technology [65]. Unless the organization gains confidence in social networks, the use of social media in the management process will have no influence in terms of improving the company’s performance [66]. Trusted social networks enable businesses to receive the best prices and increase sales, thus enhancing the performance of the firm [66]. Social networks serve as a link between internal and external globalization and business performance [67]. The authors of this paper formulated the following hypothesis based on past research findings:

**H7:** *Digital transformation has a positive effect on hospital performance mediated by networking capability*.

### 2.10. Resource Integration, Hospital Performance, and Networking Capability

Networking capabilities can assist businesses in overcoming constrained resources, which frequently impede corporate expansion and disturb company performance [68]. One intangible resource, market knowledge, is an in-depth grasp not just on a company’s decisions but also of the arrangements in which these decisions will be made. The association between market knowledge and performance is mediated by networking skills [69]. Based on past research findings, the authors propose the following hypothesis:

**H8:** *Resource integration has a positive effect on hospital performance mediated by networking capabilities*.

## 3. Methods

This study is cross-sectional and non-experimental. Cross-sectional design is used to describe a phenomenon and explain its link to other phenomena during data collection. Cross-sectional research collects all data at once to answer research questions [70]. The closed-ended questionnaire consisting of items on informed consent, demographic data, hospital performance questionnaires [71], digital transformation questionnaires [72,73], resource integration questionnaires, networking capability questionnaires [29,52], and medical tourism ecosystem questionnaires [25] was sent by email and/or Whatsapp to Directors of Class A and B hospitals scattered all over the big island in Indonesia such as in Java, Sumatra, Bali, Kalimantan, Papua, and Sulawesi. Participants were determined by using a method of data collection known as simple random sampling. Data collection was carried out over three months. Using the analysis provided by the structural equation model, the data are processed in order to evaluate the hypothesis. After the study, the researchers will provide recommendations.

### 3.1. Sample

An entity or person selected at random from the population constitutes the sample [70]. The research data were analyzed using a structural equation model. According to Loehlin and Beaujean [74], the minimum number of samples required for research that uses a structural equation model is 200. According to Regulation No. 76 of the Ministry of Health of the Republic of Indonesia from 2015, hospitals offering medical tourism services must be classified as either “Class A” or “Class B” and must hold national plenary accreditation. There are a total of 374 hospitals included in this analysis, including 50 class A hospitals and 324 class B hospitals that have earned full accreditation from both KARS in 2021 and JCI.

Table 1 shows that the sample size for this study was 200 hospitals, with 27 being classified as “A” and 173 as “B.” There was 1 director from the board of directors at each of the hospitals in the sample. The hospitals will be selected using a simple random sampling method.

### 3.2. Operational Variables

Some of the operational variables employed in this study described in Table 2.

### 3.3. Survey Development

All of the questions on the research questionnaire were closed and measured on a Likert scale. To gauge the extent to which respondents agree or disagree with a proposition, the Likert scale employs five levels of agreement or disagreement. It goes from “strongly disagree” (1) to “disagree” (2), “neutral” (3), “agree” (4), and “strongly agree” (5) [70]. Once we have collected enough information, we can analyze it to see how each variable we evaluated is related to the others.

Before using the questionnaire to verify the hypothesis, the researcher first verified its validity and reliability to ensure that it met the criterion of excellent data [75]. Validity test was carried out using a Confirmatory Factor Analysis (CFA) with LISREL. While the reliability test is seen from two measurements, namely, construct reliability (CR) and variance extracted (VE), the expected CR value is > 0.70 (reliable) and the expected VE value is > 0.50. The validity and reliability tests conducted on the research questionnaire indicate that all research indicators were valid and reliable because the CR value is more than 0.7 and the VE value is greater than 0.5. 

## 4. Results

### 4.1. Profile of the Respondents

Table 3 reveals that of the 241 hospitals classified as either class A or class B, 86.3% had a president director or hospital director. In 9.5% of the institutions included in this analysis, a director or deputy director of medical services was present. A total of 4.1% of the hospitals surveyed had a director or deputy director of medical support on staff. This shows that most of the hospitals in this study are either class A or B and were fully accredited at the national and/or international levels, as shown by the presence of a president director/hospital director.

### 4.2. Description of Variables

Table 4 shows the means of the variables for each indicator in this research. Interpretation of the average or mean value is used to see an overview of respondents’ perceptions of questions or indicators.

### 4.3. Measurements Model Analysis

#### 4.3.1. Testing the Validity and Reliability of the Research Variables Construct (Model of Second Confirmatory Factor Analysis)

The SEM model is investigated using two approaches in this study: first, the model parameters are determined using the Confirmatory Factor Analysis (CFA) measurement model, and then the structural model is tested. This study adopts a Confirmatory Factor Analysis (CFA) measurement model in which each research variable is measured using a number of different indicators. Consequently, the initial step of CFA analysis is simplifying each measurement model by obtaining a latent variable score for each dimension. In Lisrel 8.80, the standardized loading factor (SLF) value is available in the entirely standardize solution output, whereas the t-value is available in the output measurement equation. Two metrics are used to determine the reliability test: construct reliability (CR) and variance removed (VE). The expected CR value exceeds 0.70 (reliable), while the expected VE value exceeds 0.50.

#### 4.3.2. Overall Model Fit

To evaluate the degree of fit or goodness of fit (GOF) between the data and the model, one can check the SEM (Overall Model Fit) model’s precision. A good model of the estimation data findings is one that satisfies (good) the model fit requirements. Godness of Fit measurements in this study are described in Table 5 below:

All GOF (analysis of the overall fit of the model) values in Table 5 show Good Fit; hence, the model’s overall fit is satisfactory (Model Fit).

#### 4.3.3. Manifest Variables Analysis

Table 6 below is a summary of the manifest variables (dimensions) that make up the latent variables.

As can be seen in the table above, the manifest used to construct the latent variable can be clearly interpreted if the CR value is more than 0.6 and the VE value is greater than 0.5. In other words, it exceeds the benchmark on the basis of the combined values of CR and VE for all variables. Since the latent variable is made up of the manifest variables, this means that all of them are good.

### 4.4. Hypothesis Testing and Discussion

#### 4.4.1. Hypothesis Test

The research model is depicted in Figure 1 below:

The hypothesis of causation between variables is evaluated after the variable construct has been examined. Table 7 below shows the results of the hypothesis testing that was carried out. 

#### 4.4.2. Hypothesis 1: Digital Transformation has a Positive Effect on Hospital Performance

The standard coefficient value of the digital transformation variable on hospital performance is 0.195 and has a positive sign, according to Table 7. This indicates that digital transformation positively impacts hospital performance. Each digital transformation unit added to a hospital will boost its performance by 0.195.

The purpose of testing hypothesis 1 us to find out whether there is a positive effect of digital transformation on hospital performance. The null hypothesis (H0) and the alternative hypothesis (Ha) are given below:**H0:** *There is no positive effect of digital transformation on hospital performance*;**Ha:***There is a positive influence of digital transformation on hospital performance*.

From the results of hypothesis testing, it was determined that the t-value (0.863) was less than 1.96; hence, Ha was **rejected** and H0 was not. This explains why the impact of digital transformation on hospital performance is favorable but not statistically significant.

The results of this study are different from those from research conducted by Bharadwaj et al. [40], Nwankpa et al. [39], and Wang et al. [46] in which digital transformation was shown to positively affect company performance. The development of health technology at competitive prices can be one of the success factors for South Korea in developing the medical tourism industry [76]. Meanwhile, Bulatovic and Iankova [10] stated that one of the obstacles in the medical tourism industry in the UAE is the high cost of services. This shows that hospitals not only need to have digital technology but must be able to take advantage of the technology they have and offer competitive prices for the services provided.

#### 4.4.3. Hypothesis 2: Digital Transformation has a Positive Effect on Networking Capability

According to Table 7, the standard coefficient of the digital transformation variable on network capacity is 0.326 and positive. This indicates that digital transformation is positively associated with networking capability (KB). Each additional unit of digital transformation improves networking capacity by 0.326.

Testing hypothesis 2 aims to find out whether there is a positive effect of digital transformation on networking capability. The null hypothesis (H0) and the alternative hypothesis (Ha) are given below:**H0:** *There is no positive effect of digital transformation on networking capability*;**Ha:** *There is a positive influence of digital transformation on networking capability*.

The results of hypothesis testing revealed that the t-value (2.094) was >1.96, indicating that Ha was **accepted** and H0 was rejected. This explains why digital transformation has a significant and positive impact on networking capability.

These results are in accordance with the research of Cenamor et al. [49], which states that digital platform capability has a significant positive effect on networking capabilities. Zoppelletto et al. [77] also stated that Business Network Commons can be supported by adopting a digital transformation strategy. Digital transformation is able to create networking opportunities for companies [78]. Digital transformation in health services changes the value creation mechanism by expanding networks and collaboration with stakeholders (e.g., digital companies) in the industry [79]. Digital transformation not only changes the mechanism between actors in health services but revises the overall health service landscape with the involvement of new market actors/players [80].

#### 4.4.4. Hypothesis 3: Networking Capability has a Positive Effect on Hospital Performance

Data from Table 7 shows that there is a beneficial effect of hospitals’ networking capabilities on patient outcomes. It follows that a hospital’s efficiency improves with each increased KB of networking capacity. The efficiency of a hospital improves by 0.312 percentage points for every additional kilobyte of network capacity.

The goal of hypothesis 3 is to find out how a hospital’s networking capabilities improves patient care. In the following graphic, we contrast a null hypothesis (H0) with an alternative hypothesis (Ha):**H0:***There is no positive effect of networking capability on hospital performance*;**Ha:** *There is a positive influence of networking capability on hospital performance*.

The t-value (3.887) was greater than 1.96 as a result of hypothesis testing, so Ha was **accepted** and H0 was rejected. This explains why networking capability has a significant and positive impact on hospital performance.

These results are consistent with research by Yang et al. [51] and Kurniawan [52] which state that networking capabilities improve company performance. In the hospitality industry, networking capabilities also affect the company’s strategic performance [12]. In health services, absorptive capacity, which is a dimension of networking ability, also influences hospital performance [50]. Collaboration from hospitals, insurance, embassies, and other businesses is a factor of administrative service quality expected by patients in the medical tourism industry [81]. Network collaboration makes patients satisfied with medical tourism services, which is a component of non-financial performance.

#### 4.4.5. Hypothesis 4: Resource Integration has a Positive Effect on Hospital Performance

The standard coefficient variable value of resource integration on hospital performance is 4.156 and has a positive sign, according to Table 7. This indicates a positive association between resource integration and hospital performance. Each resource integration unit added to a hospital will boost its performance by 4.156.

While testing hypothesis 4, the effect of resource integration on hospital performance is investigated. The null hypothesis (H0) and the alternative hypothesis (Ha) are given below:**H0:** *There is no positive effect of resource integration and hospital performance*;**Ha:** *There is a positive influence of resource integration and hospital performance*.

The results of hypothesis testing revealed that the t-value (2164) was greater than 1.96; hence, Ha was **accepted** and H0 was rejected. This describes how resource integration has a positive and substantial impact on hospital performance.

These results are consistent with the research of Barney and Clark [55] which states that companies that focus on resources and pay attention to the characteristics of the resources they have can improve company performance. Meng and Yang [58] also mention that resource integration can improve company performance, while Fang et al. [56] stated that company performance can be predicted from company resources. In the medical tourism industry, the integration of resources, especially social (intangible) resources, has a major impact on individual decisions to undertake medical tourism [82]. Resources such as the competence of medical staff, service quality, and communication are important factors that affect patient loyalty in medical tourism and thus improve hospital performance [83].

#### 4.4.6. Hypothesis 5: Resource Integration has a Positive Effect on Networking Capability

According to Table 7, the standard coefficient variable of resources integration on networking capability has a value of 1.018 and a positive sign. This indicates a positive association between resource integration and networking capability. Each resources integration unit added will boost networking capability by 1.018.

Testing hypothesis 5 is designed to determine the favorable impact of resources integration on networking capability. The null hypothesis (H0) and the alternative hypothesis (Ha) are given below:**H0:** *There is no positive effect of resource integration on networking ability*;**Ha:** *There is a positive influence of resource integration on networking ability*.

Based on the findings of the hypothesis testing, the t-value (6.045) was more than 1.96; hence, Ha was **accepted** and H0 was rejected. This illustrates how resources integration has a positive and substantial effect on networking capabilities.

These results are consistent with the research of Vesalainen and Halaka [47], which states that company resources are a manifestation of customer-oriented networking capabilities. Networking capabilities are the right strategy for companies with limited resources to meet their resource needs and share business risks [60,61,62]. Bao and Hua [84] stated that in the tourism industry it is necessary to first identify the resources they have, where the resources are obtained, the utilization of resources and the development of tourism programs before finally building a network.

#### 4.4.7. Hypothesis 6: The Medical Tourism Ecosystem Strengthens the Relationship between Networking Capability and Hospital Performance

According to Table 7, the value of the standard coefficient variable in strengthening the relationship between networking capability and hospital performance is 0.0132 and is positive.

Meanwhile, hypothesis 6 seeks to determine whether the medical tourism ecosystem can improve the relationship between networking capability (KB) and hospital performance. The null hypothesis (H0) and the alternative hypothesis (Ha) are shown below:**H0:** *The medical tourism ecosystem cannot strengthen the relationship between networking capability and hospital performance*;**Ha:** *The medical tourism ecosystem can strengthen the relationship between networking capability and hospital performance*.

According to the results of hypothesis testing, the t-value (2.082) was >1.96, so Ha was **accepted** and H0 was rejected. This explains how the medical tourism ecosystem can improve the link between networking capability and hospital performance.

The successful development of medical tourism is highly dependent on good partnerships between all stakeholders in the medical tourism ecosystem [38]. Bulatovic and Iankova [10] mentioned that a network of more efficient medical care suppliers, tourism suppliers, the government, and intermediaries must be formed to ensure the growth of the medical tourism industry. Vovk et al. [85] also stated that developing medical tourism depends on healthcare funding, the development of political and civil freedoms, infrastructure and institutional factors formed under the influence of national socio-economic policies, and government cooperation with businesses and households under financing activities included in the medical tourism ecosystem. Park et al. [86] added that the presence of facilitating agents in the medical tourism ecosystem moderates the effects of hospitals’ service quality dimensions on service satisfaction.

#### 4.4.8. Hypothesis 7: Digital Transformation has a Positive Effect on Hospital Performance Mediated by Networking Capability

According to Table 7, the standard coefficient variable digital transformation on hospital performance mediated by networking capability is 0.102, which is positive.

Hypothesis testing 7 seeks to ascertain the impact of digital transformation on hospital performance as mediated by networking capability. The null hypothesis (H0) and the alternative hypothesis (Ha) are shown below:**H0:** *Digital transformation has no positive effect on hospital performance mediated by networking capability*;**Ha:** *Digital transformation has a positive effect on hospital performance mediated by networking capability*.

According to the results of hypothesis testing, the t-value (1.693) was <1.96, so Ha was **rejected** and H0 failed to be rejected. As a result, networking capability cannot mitigate the impact of digital transformation on hospital performance.

The results of this study are different from those of Schallmo and Williams [87], who state that company performance is influenced by technology and the connectivity of all stakeholders throughout the market chain. Yang et al. [51] also have a different opinion from the results of this study, namely, networking skills strengthen the relationship between digital capabilities and company performance growth. A different opinion was also expressed by Herrmann et al. [88], who stated that digital transformation in healthcare is an opportunity to improve performance by collaborating with other companies.

#### 4.4.9. Hypothesis 8: Resource Integration has a Positive Effect on Hospital Performance Mediated by Networking Capabilities

The value of the standard coefficient variable integration of resources on hospital performance is mediated by networking ability and has a positive sign, as shown in Table 7.

While testing hypothesis 8, the effect of resources integration on hospital performance as mediated by networking capabilities will be determined. The null hypothesis (H0) and the alternative hypothesis (Ha) are given below:**H0:** *Resource integration has no positive effect on hospital performance, which is mediated by networking skills*;**Ha:** *Resource integration has a positive effect on hospital performance, which is mediated by networking capabilities*.

From the results of testing the hypothesis, it was found that the t-value (2.988) was >1.96, so Ha was **accepted** and H0 was rejected. This demonstrates that networking capabilities can mediate the effect of resource integration on hospital performance.

These results are in accordance with the research of Lu and Beamish [68], which showed that networking capabilities can help companies overcome limited resources, which often hinder company expansion, thereby disrupting company performance. Networking skills mediate the relationship between market knowledge, which is one of the market’s intangible resources, and performance [69].

### 4.5. Implications

This study supports the stakeholder theory. The results of this study indicate that the medical tourism ecosystem can strengthen the relationship between networking capabilities and hospital performance. The existence of a medical tourism ecosystem is able to help health service providers and the tourism industry to provide mutual benefits in improving company performance. The Ministry of Tourism and Creative Economy and the Ministry of Health must work together to form an Indonesian Health Tourism Board so that all stakeholders can come together under one ecosystem called the medical tourism ecosystem. With the existence of the Indonesian Health Tourism Board, organizations can strengthen their relationships with external groups and develop competitive advantages tailored to them. The Indonesian Health Tourism Board also functions as a communication forum for all stakeholders to help hospitals establish cooperation outside the health industry (e.g., travel agents, insurance, etc.) because establishing relationships with other medical tourism stakeholders such as travel agents and accommodation is also crucial to develop medical tourism industry.

This research has practical applications, such as the potential for facilitating digital transformation to enhance hospital business procedures, particularly in medical tourism programs. Based on descriptive statistics, hospitals have not fully adopted digital technology (mean 3.39), so hospitals must carry out digital transformation as a whole in both technological and non-technological dimensions. Hospitals must also consider the price of the services offered despite the advanced medical technology provided in order to be competitive. The role of the Ministry of Health in supporting the digital transformation of hospitals in Indonesia is to create a national digital transformation road map. In addition, the Ministry of Finance needs to reduce the tax rate on advanced medical devices so that hospitals can offer more affordable and competitive service prices.

Furthermore, hospitals are expected to increase resources integration, particularly intangible resources such as skilled resource availability and effective staff communication skills because, based on descriptive statistic, the availability of employees with good communication skills (3.9) has the lowest mean of all resource indicators used in this study. Hospitals and professional organizations need to conduct effective communication training for all employees and care-giving professionals.

The outcomes of the study also indicate that hospitals create and inform clients about the superior services and comprehensive amenities they offer. The Ministry of Tourism and Creative Economy and the Ministry of Health must help socialize the superior services of the hospital in Indonesia because, based on descriptive statistics, the regional reputations of hospitals (2.9) have the lowest mean of all hospital performance indicators used.

## 5. Conclusions

This study concludes that digital transformation and resource integration have a favorable influence on networking capacities based on hypothesis testing. Hospitals need to prepare their resources as their main capital in developing networking capabilities. Digital transformation, on the other hand, has no positive impact on hospital performance. According to the findings of the study, digital transformation has a good link with hospital performance but has no statistically significant effect on hospital performance. Meanwhile, resource integration and networking capabilities improve hospital performance. Hospitals need to carry out digital transformation and have human resources capable of adapting to digital transformation to improve hospital performance through networking capabilities. Similarly, a medical tourism ecosystem would improve the link between networking skills and hospital performance. The findings of this study are likely to serve as a resource for policymakers and the health industry as they develop services for the tourism market.

## Figures and Tables

**Figure 1 ijerph-20-00374-f001:**
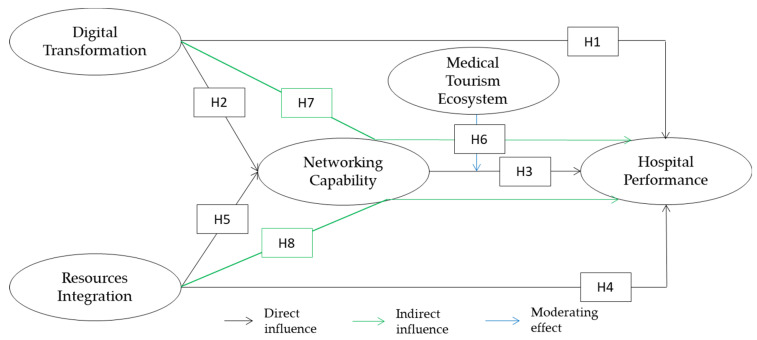
Research Model.

**Table 1 ijerph-20-00374-t001:** Determination of Respondent’s Proportions.

No	Hospital Class	Number of Hospitals	Sample	Subtotal	Integration
1	Hospital Class A	50	50 (200/374)	26.7	27
2	Hospital Class B	324	324 (200/374)	173.2	173
Total		374			200

**Table 2 ijerph-20-00374-t002:** Operational Variables.

Hospital Performance	Definition	Ability to achieve hospital goals		
	Dimension	Code	Indicator	Scale
	Financial	F_1	Increase in hospital income	Interval
		F_2	Increased net profit/residual operating results of the hospital	Interval
		F_3	Hospital shows management improvement	Interval
	Non- Financial	NF_1	Increased patient satisfaction rate	Interval
		NF_2	Decreased medical procedure error rate	Interval
		NF_3	Hospital reputation nationally	Interval
		NF_4	Regional Reputation of Hospital	Interval
		NF_5	Increased medical tourism patient visits	Interval
		NF_6	Hospital growth	Interval
Networking Capability	Definition	Ability to locate, manage, and leverage networks to create value		
	Dimension	Code	Indicator	Scale
	Looking for partners	MM_1	Systems and mechanisms for identifying partners	Interval
		MM_2	Systems and mechanisms for selecting partners	Interval
		MM_3	Systems and mechanisms to find partners under certain conditions	Interval
	Manage network	MJ_1	Relationship analysis with partners	Interval
		MJ_2	Partnership relationship adjustment	Interval
		MJ_3	Integration of network activities into business operational processes	Interval
	Take advantage of the network	MFJ_1	Accurately receive help from partners	Interval
		MFJ_2	Receive help from partners at the right time	Interval
		MFJ_3	Partners refer to third parties if they cannot provide direct assistance	Interval
		MFJ_4	Professional inter-collaboration with other hospitals	Interval
Resource Integration	Definition	The process of integrating resources to create value		
	Dimension	Code	Indicator	Scale
	Tangible	TN_1	The hospital’s location is easily accessible	Interval
		TN_2	Attractive hospital structures and designs	Interval
		TN_3	Medical equipment with cutting-edge technology is readily available.	Interval
	Intangible	ITN_1	In the view of patients, the hospital’s reputation is excellent.	Interval
		ITN_2	Medical personnel who are skilled in their specialty are available.	Interval
		ITN_3	Availability of employees with good communication skills.	Interval
		ITN_4	The ability of the hospital to integrate resources to improve performance.	Interval
Digital Transformation	Definition	Using technology to improve performance and create value		
	Dimension	Code	Indicator	Scale
	Technology	T_1	Complete adoption of digital technology	Interval
		T_2	Use of digital technology to improve service activities	Interval
		T_3	Digital technology updates in accordance with the latest developments in the health sector	Interval
	Non Technology	NT_1	Staff support for changes according to digital transformation needs	Interval
		NT_2	Patient involvement in service processes that use digital technology	Interval
		NT_3	Adjustment of business processes to the development of digital technology	Interval
Medical Tourism Ecosystem	Definition	Stakeholders involved in medical tourism		
		Code	Indicator	Scale
		EWM_1	The establishment of the Indonesia Health Tourism Board	Interval
		EWM_2	Government support in the advancement of medical tourism	Interval
		EWM_3	Support of tourism stakeholders in the medical tourism industry	Interval
		EWM_4	Support of medical stakeholders in the medical tourism industry	Interval
		EWM_5	Ecosystem benefits to stakeholders in the medical tourism industry	Interval
		EWM_6	The rules of the medical tourism ecosystem	Interval

**Table 3 ijerph-20-00374-t003:** Profile of Respondents based on Position.

Respondent Position	Frequency	Percentage
President Director/Hospital Director	208	86.3%
Director of Medical Services/Deputy Director of Medical Services	23	9.5%
Medical Support Director/Deputy Medical Support Director	10	4.1%
Total	241	100%

**Table 4 ijerph-20-00374-t004:** Description of Variables.

Hospital Performance	Dimension	Code	Indicator	Mean
	Financial	F_1	Increase in hospital income	4
		F_2	Increased net profit/residual operating results Hospital	3.93
		F_3	Hospital shows management improvement	4.06
	Means of financial dimension			4
	Non- Financial	NF_1	Increased patient satisfaction rate	4.14
		NF_2	Decreased medical procedure error rate	3.73
		NF_3	Hospital reputation nationally	4.03
		NF_4	Regional Reputation of Hospital	2.9
		NF_5	Increased medical tourism patient visits	4.13
		NF_6	Hospital Growth	4.17
	Means of non-financial dimension			3.85
	Means of hospital performance			3.9
Networking Capability	Dimension	Code	Indicator	Mean
	Looking for partners	MM_1	Systems and mechanisms for identifying partners	3.75
		MM_2	Systems and mechanisms for selecting partners	3.78
		MM_3	Systems and mechanisms to find partners under certain conditions	3.79
	Means of looking for partners dimension			3.77
	Manage network	MJ_1	Relationship analysis with partners	3.89
		MJ_2	Partnership relationship adjustment	3.99
		MJ_3	Integration of network activities into business operational processes	3.92
	Means of management network dimension			3.93
	Take advantage of the network	MFJ_1	Accurately receive help from partners	3.78
		MFJ_2	Receives help from partners at the right time	3.81
		MFJ_3	Partners refer to third parties if they cannot provide direct assistance	3.89
		MFJ_4	Professional inter-collaboration with other hospitals	4.04
	Means of take advantage of the network dimension			3.88
	Means of networking capability			3.86
Resource Integration	Dimension	Code	Indicator	Mean
	Tangible	TN_1	The hospital’s location is easily accessible	4.57
		TN_2	Attractive hospital structures and designs	4.04
		TN_3	Medical equipment with cutting-edge technology is readily available.	3.97
	Mean of tangible dimension			4.19
	Intangible	ITN_1	In the view of patients, the hospital’s reputation is excellent.	4.20
		ITN_2	Medical personnel who are skilled in their specialty are available.	4.38
		ITN_3	Availability of employees with good communication skills.	3.90
		ITN_4	The ability of the hospital to integrate resources to improve performance.	4.01
	Means of intangible dimension			4.12
	Means of resources integration			4.16
Digital Transformation	Dimension	Code	Indicator	Mean
	Technology	T_1	Complete adoption of digital technology	3.39
		T_2	Use of digital technology to improve service activities	3.45
		T_3	Digital technology updates in accordance with the latest developments in the health sector	3.95
	Means of technology dimension			3.6
	Non- Technology	NT_1	Staff support for changes according to digital transformation needs	4.1
		NT_2	Patient involvement in service processes that use digital technology	3.78
		NT_3	Adjustment of business processes to the development of digital technology	4.03
	Means of non-technology dimension			3.97
	Means of digital transformation			3.78
Medical Tourism Ecosystem	Code	Indicator		Mean
	EWM_1	The establishment of the Indonesia Health Tourism Board		4.12
	EWM_2	Government support in the advancement of medical tourism		3.83
	EWM_3	Support of tourism stakeholders in the medical tourism industry		3.81
	EWM_4	Support of medical stakeholders in the medical tourism industry		3.76
	EWM_5	Ecosystem benefits to stakeholders in the medical tourism industry		3.73
	EWM_6	The rules of the medical tourism ecosystem		3.04
	Means of medical tourism ecosystem			3.71

**Table 5 ijerph-20-00374-t005:** Measurement Goodness of Fit Statistic Structural Model.

Indicators	Compute GoFI Value	Standard Value for Good Fit	Conclusion
*p*-value	0.0	>0.05	Marginal value
RMSEA	0.07	<0.08	Good Fit
NFI	0.92	>0.9	Good Fit
NNFI	0.93	>0.9	Good Fit
CFI	0.94	>0.9	Good Fit
IFI	0.94	>0.9	Good Fit
GFI	0.93	>0.9	Good Fit

**Table 6 ijerph-20-00374-t006:** Manifest Variables Analysis.

Variables	Manifest Variables	t-Value	Standardize Loading Factors	Composite Reliability (CR)	Variance Extracted (VE)
Digital transformation	Technology	15.85	0.83	0.8	0.67
	Non Technology	15.78	0.83		
Resources integration	Tangible	11.23	0.68	0.7	0.52
	Intangible	12.84	0.75		
Networking capability	Looking for partners	***	0.76	0.83	0.62
	Manage network	15.14	0.8		
	Take advantage of the network	15.13	0.8		
Hospital performance	Financial	***	1.3	0.78	0.65
	Non-financial	7.4	1.43		
Medical Tourism ecosystem	EWM _1	13.11	0.75	0.91	0.64
	EWM _2	15.40	0.81		
	EWM _3	17.30	0.83		
	EWM _4	16.50	0.81		
	EWM _5	16.35	0.81		
	EWM _6	9.63	0.78		

*** There is no path in the structural model path diagram (t-value estimation). This is because the variable has been set to be the variance reference, which means that the indicator variable is significantly related to the latent variable.

**Table 7 ijerph-20-00374-t007:** Hypothesis Test Results.

No	Hypothesis	t-Value	Standardize Coefficient	Conclusion
H1	Digital transformation has a positive effect on hospital performance	0.863	0.195	Hypothesis rejected
H2	Digital transformation has a positive effect on networking capabilities	2.094	0.326	Hypothesis accepted
H3	Networking capabilities has a positive effect on hospital performance	3.877	0.312	Hypothesis accepted
H4	Resources integration has a positive effect on hospital performance	2.164	4.156	Hypothesis accepted
H5	Resources integration has a positive effect on networking capabilities	6.045	1.018	Hypothesis accepted
H6	Medical tourism ecosystem strengthens the relationship between networking capability and hospital performance	2.082	0.132	Hypothesis accepted
H7	Digital transformation has a positive effect on hospital performance mediated by networking capabilities	1.693	0.102	Hypothesis rejected
H8	Resources integration has a positive effect on hospital performance mediated by networking capabilities	2.988	0.318	Hypothesis accepted

## Data Availability

Not applicable.

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
