# Peer review of "Interplay between Networking Capability and Hospital Performance in Indonesia’s Medical Tourism Sector"

_ijerph, 2022, doi:10.3390/ijerph20010374_

Round 1

Reviewer 1 Report

2022 12 19  ijerph 2119441

Digital transformation had changed the world, including the medical tourism sector. The research studied on the interplay between networking capacity and hospital performance in Indonesia medical tourism sector. The structure equation model was applied in this study, and the authors proposed eight hypothesis tests to investigate the interplay of the key elements (hospital performance, networking capability, resource integration, digital transformation, medical tourism ecosystem). Lisrel 8.8 was the software used for the analysis.

1. The current quality and the services offered in Indonesia is unknown and not offered in the introduction. Through this study, there might be some further regarded information, but failed to offered. There the Reviewer suggested the authors that (a) add more information about the variables and the current quality and current services of the medical tourism services in Indonesia; (2) add the descriptive statistics of the variables. These information needs to be clearly offered to improving the understanding of the readers regarding this sector in the study site.

2. In Table 2, the operational variables are defined and the scale is determined. However, the statistics of the variable is invisible. In the column of “scale” in Table 2, can also offered the mode or the medium of the variables as the respond of the respondents.  

3. In page 2, three tiers of stakeholder within the ecosystem of the medical tourism sector are addressed. It is suggested that the regarding stakeholders that related to the variables under study in this study ought to be discussed and further addressed, on the purpose to better manage the sectors with the benefit of the stakeholders.

4. Why the paper choose to review the five countries in page 1? The five countries do not include the one under the present investigation. What implication can be drawn from this literature for the reference of the present paper.  

5. In line 47-63 in page 2, the medical sector is introduced, but we cannot realize what is the significance of this study to investigate the medical tourism sectors in Indonesian. Please offer more information to highlight the purpose of this investigation. The information of the studies countries needs to be offered, such as medical level, international ranking, ranking among neighboring countries, competitive advantage, attractiveness, and so on.

6. Sector “4.3.2” repeated appears inappropriately.  And the null hypothesis should appear before alternative hypothesis, in general.

7. “…are given below (H0).”in P11-P14, seems inappropriate. The H0 in parentheses needs to be removed.

8. The implications should be addressed more, regarding the statistics of variables and the study results to offer insight information to better manage the sector of medical tourism, as well as the medical hospitals in the study sites.

9. The research design, the methodology and the study results should be offered in detailed information.      

Reviewer 2 Report

Thanks for researching the relevant topic for the researchers and industry (medicine and tourism) on the Medical tourism and digitalization issue. 

The technical suggestion is to convert rupiahs to US dollars. It will support readers' understanding of the volume of money. 

The advice is to add information on research methods (ch.3), including a period of obtaining data from the survey etc.

Please, strengthen the conclusion part. 
